# The Impact of Artificial Intelligence Technology Stimuli on Sustainable Consumption Behavior: Evidence from Ant Forest Users in China

**DOI:** 10.3390/bs13070604

**Published:** 2023-07-20

**Authors:** Ping Cao, Shuailong Liu

**Affiliations:** Shanghai Institute of Technology, College of Economics and Management, Shanghai 200235, China

**Keywords:** sustainable consumption behavior, artificial intelligence technology stimuli, customer-perceived value, customer stickiness, Ant Forest

## Abstract

With the global economy and population growing rapidly, the problems of excessive resource consumption and environmental pollution have become increasingly serious. Thus, the need to promote sustainable development has become more urgent. Sustainable consumption behavior plays a crucial role in achieving sustainable development goals as it can significantly reduce both greenhouse gas emissions and resource consumption. Artificial intelligence technology has broken the limitations of time and space in environmental protection. For example, the Ant Forest leverages the design of “green energy” to inspire the public to engage in energy-saving and emission-reducing activities. To examine the impact mechanisms of customers’ sustainable consumption behavior, this study applies the stimulus-organism-response theory and the theory of planned behavior. The study conducts regression analysis and bootstrapping methods on a sample consisting of 280 Ant Forest users to explore the influence of artificial intelligence technology stimuli on sustainable consumption behavior and the mediating effects of customer-perceived value and customer stickiness. The results demonstrate a “linkage effect” between online green consumption habits and offline sustainable consumption behavior. Moreover, the study finds that passion and usability indirectly promote offline sustainable consumption behavior through customer-perceived value and customer stickiness. Specifically, the influence of customer-perceived emotional value (β = 0.121; β = 0.100) is stronger than that of customer-perceived social value (β = 0.043; β = 0.038). Due to the limitation of the sample size, future research should broaden its scope by incorporating additional variables, specifically customer-specific factors. Furthermore, more advanced research methods, such as big data analysis, should be employed to comprehensively explore the influencing factors of sustainable consumer behavior.

## 1. Introduction

With the constant improvement of people’s living standards and the continuous growth of consumption demand, the consumption of natural resources has far exceeded its regenerative capacity [1]. In this context, sustainable development has gradually become a focus of scholars’ attention, and people are increasingly aware of the importance of consumption behavior for environmental protection [2]. Sustainable consumption behavior is a green, ecological, and moderate consumption method that emphasizes minimizing the impact on the environment while also not affecting people’s quality of life [3]. This consumption pattern is based on the concepts of circular economy, low-carbon environmental protection, and responsibility to society. It encourages people to opt for reusable products and renewable resources and adopt simple, environmentally friendly lifestyles. Not only does this pattern reduce energy and resource consumption, but it also helps to protect the environment, mitigate climate change, and improve the living conditions of humans and animals on Earth. While sustainable consumption behavior is critical, studies indicate that most behavior in this regard is not sustainable. Thus, it is significant to study the factors that influence sustainability in order to encourage sustainable consumption.

Historically, research on sustainable consumption behavior has focused mainly on internal psychological and individual factors, including cognition, emotions, attitudes, beliefs, and behavior patterns [4,5,6]. There is a lack of studies that explore external factors, with the majority adopting institutional perspectives such as the government and businesses [7,8]. Consequently, the impact of external environmental variables, such as the stimulation provided by information technology, has not received sufficient attention. With the rapid development of internet technology, network information inundates every aspect of people’s lives. For example, some advertisements and promotional activities convey green, environmental, and sustainable consumer concepts to consumers, thus influencing their purchasing behavior. Artificial intelligence (AI) technology is more widely used in this regard. For instance, intelligent customer service robots, equipped with technologies such as speech recognition, natural language processing, and chatbots, can assist consumers in resolving problems in a timely manner and improve their shopping experience. Additionally, AI technology can be employed to develop mobile applications that gamify green consumer or sustainable behavior to the public.

Ant Group, for instance, launched its Ant Forest mobile application built with big data and AI technology in 2016. The application’s objective is to prompt the public in practicing sustainable consumption behavior while also addressing climate change and environmental issues. The application encourages the public to walk instead of driving and to purchase tickets online to reduce carbon emissions. The carbon saved is calculated by means of virtual computing and transformed into “green energy” to plant virtual trees within the application. When these virtual trees grow, Ant Forest plants a real one on the earth to cultivate and incentivize low-carbon environmental behaviors. Furthermore, Ant Forest has incorporated diverse interactive elements, such as energy theft, watering assistance, co-planting, energy competitions, and leaderboards, among others, to enhance the immersive gaming experience and foster emotional communication [9]. Figure 1 illustrates some application functionalities. In this way, effective promotion of green consumption habits can be achieved, and consumer behaviors towards sustainable consumption can be guided. The Ant Forest has garnered substantial involvement from a large number of Chinese citizens and, owing to its significant contributions to environmental conservation, received the esteemed United Nations’ “Earth Champion Award” on 26 September 2019. According to Ant Group’s 2022 Sustainable Development Report, there are more than 650 million users in “Ant Forest” who have embraced a green and low-carbon lifestyle. As a result, they have collectively generated over 26 million tons of “green energy”. These users have successfully planted 400 million trees and protected a social welfare area of 2 million hectares, with the help of applications and funding provided by Ant Group. Despite the noteworthy accomplishments of Ant Forest, it still requires empirical evidence to determine if its users are able to uphold sustainable consumption practices offline, comparable to their behaviors online. Hence, it is crucial to ascertain the potential influence of Ant Forest users’ online environmental behaviors on their offline sustainable consumption practices.

According to the Stimuli-Organism-Response (S-O-R) theory, when organic entities are exposed to external stimuli generated by AI technology, their perceived value is enhanced. When these entities perceive this stimulus, they respond accordingly. In the process of interacting with Ant Forest, the public can experience the passion and usability brought by it, which helps to strengthen their perceived emotional and social value, attracting customers to use it continuously, and resulting in customer stickiness. According to the Theory of Planned Behavior (TPB), an individual’s intentions affect their consumption behavior. Studies have shown that there is a cross-domain linkage phenomenon between online and offline sustainable behaviors, and consumers who often participate in online environmental activities are more likely to engage in environmentally-friendly behaviors in their daily lives. It is clear that the stimulation of online artificial intelligence technology may trigger offline sustainable behavior, while offline environmental behaviors can also encourage the public to actively participate in online green public welfare activities, forming a closed loop of online and offline linkage effects. This study aims to investigate the influence of external environmental variables on offline consumption behavior, and this study integrates the SOR theory with the TPB theory for the first time, exploring the influence mechanism of customer-sustainable consumption behavior from a psychological standpoint. The theoretical framework and development of hypotheses will be outlined in the upcoming section. A detailed description of the questionnaire design and data analysis methods will be presented in Section 3. Section 4 will conduct a comprehensive analysis of the empirical results. Lastly, the concluding chapter will explore the implications, limitations, and future research directions derived from the findings of this study.

## 2. Theoretical Foundation and Hypotheses

### 2.1. Theoretical Foundation

This study is based on the S-O-R theory and the Theory of Planned Behavior. The S-O-R theory evolved from the stimulus-response (S-R) theory based on behaviorism, which explains the relationship between stimuli and responses. It argues that individual behavior is a reaction to specific stimuli, but the theory overlooks logical reasoning in the process of explaining responses. In order to compensate for this defect, Berlo and Gulley proposed the S-O-R model theory in 1957 [10]. The S-O-R theory, based on modern cognitive psychology, elucidates the impact of environmental stimuli (S) on the internal state of the organism(O), such as emotional and perceptual states, leading to various behavioral responses (R) after a series of internal processes [11,12].

The Theory of Planned Behavior (TPB) is based on the theory of reasoned action and can help us understand how people can change their behavioral patterns. The TPB theory suggests that an individual’s attitude toward behavior, subjective norms, and perceived control can affect actual behavior through behavioral intentions [13]. Since TPB can predict consumer intentions and behaviors, scholars have applied TPB to the study of consumer behavior [14]. In this study, we used Ant Forest users as data samples and constructed a conceptual model of the impact of AI technology stimuli on sustainable consumption behavior. From the perspective of the S-O-R theory and TPB theory, this study elucidates the internal mechanism of the impact of AI technology stimuli on sustainable consumption behavior.

### 2.2. Hypotheses

AI technology stimuli refer to the pleasure and practicality that customers feel when using AI technology [4]. Relevant research provides two approaches to dividing the dimensions of AI technology stimuli: hedonic and utilitarian stimulation, and perceived personalization and perceived interactivity. Hedonic and utilitarian stimulation reflect the degree to which things are pleasant and useful, while perceived personalization reflects the personalized service provided to customers through AI technology. Perceived interactivity reflects the instant connection between AI technology and customers and the ability to respond to customers’ demands. As an online application, Ant Forest encourages users to participate in energy-saving and emission-reduction activities to accumulate “green energy”. By participating in the online tree-planting activities of Ant Forest, users can plant a real tree to help the planet. The gamified way of Ant Forest brings users a pleasant experience while enhancing customers’ perception of Ant Forest’s usefulness. Therefore, this study mainly focuses on two types of stimulation, namely hedonic and utilitarian stimulation. Hedonic stimulation includes passion, which refers to the positive emotions that consumers feel in the process of interacting with AI and socializing. Utilitarian stimulation refers to usability, which refers to the extent to which AI technology is considered useful, whether it is easy to use and control.

Perceived value refers to the overall evaluation of customers’ perceived utility of a product or service, which depends on the balance between perceived gains and perceived costs [15], such as the balance between quality and price. With the development of the internet, consumer attitudes have also changed, and the dimensions of perceived value have gradually shifted from a single perspective to a comprehensive view, no longer limited to the two dimensions of perceived gains and perceived costs [16]. In gamification and interactive experience scenarios, perceived value is usually divided into four dimensions: functional value, hedonic value, social value, and emotional value [17]. Studies have shown that hedonic value and social value in gamification more effectively influence consumer behavior than functional value [18]. It is worth noting that hedonic value is included in the emotional value dimension [19]. Therefore, this study will mainly focus on the two dimensions of perceived social value and perceived emotional value. Referring to the studies of Shi et al. and Yu et al. [20,21], this study defines perceived emotional value as the emotions and feelings (such as pleasure and entertainment) evoked when using the Ant Forest application and defines perceived social value as enhancing customers’ social interaction, expanding social circles, and gaining social recognition through the use of the Ant Forest application.

Emotional value refers to the positive emotions that customers perceive when using Ant Forest, while social value refers to the social attributes that customers perceive when interacting with Ant Forest. The gamified design of Ant Forest brings a sense of pleasure to customers’ use and influences their perceived emotional value. In addition, the “stealing energy” feature of adding friends enhances the social characteristics of Ant Forest and helps improve customers’ perception of its social value. Based on the above analysis, this study views AI technology stimuli as an external environmental variable that directly impacts customers-perceived passion and usability. Therefore, this study proposes the following hypothesis:

 **H1.** *Passion* (**H1a**) *and usability* (**H1b**) *have a positive effect on customer-perceived emotional value.*

 **H2.** *Passion* (**H2a**) *and usability* (**H2b**) *have a positive effect on customer-perceived social value.*

Initial studies addressed the concept of “stickiness” primarily in relation to internet companies and websites, aiming to measure the level of user attraction based on specific characteristics [22]. With changes in consumer behavior, more scholars have studied customer stickiness from the perspective of consumers [23]. For instance, Lin emphasized that user stickiness is characterized by users’ willingness to repeatedly visit and continually utilize their preferred websites [24]. Additionally, scholars have extensively examined user stickiness in various contexts, primarily uncovering the psychological and behavioral disparities among users. Lu and Lee, for instance, focused on internet communities and argued that user stickiness is fundamentally reflected in activities such as user comments, browsing patterns, and the time that users spend engaging within the community [25]. Based on the above content and in conjunction with the research background of Ant Forest, this study defines customer stickiness as the customer’s continued willingness to use or increase their usage of Ant Forest. Presently, studies pertaining to the factors that impact customer stickiness predominantly center on the technological, customer, and external environmental dimensions. The technological dimension is primarily manifested through the technical features found on platforms or websites, including personalized recommendations and social engagement mechanisms [26], website usability, and information accessibility [27], all of which have been shown to significantly bolster customer stickiness. External environmental factors, including marketing activities and switching costs, have a significant influence on the development of customer stickiness [28]. At the customer level, customer stickiness predominantly stems from psychological factors and individual characteristics, such as customer expectations, satisfaction, and trust, all of which contribute positively to customer stickiness [29,30]. The S-O-R theory suggests that when external environmental stimuli impact an organism, the organism will produce appropriate responses. When customers experience shifts in their perceived emotional or social value towards a platform or website, it can elicit psychological preferences or rejections, which in turn affect their intention to continue or increase usage. In the case of Ant Forest, customers’ perceptions of changes in emotional and social value during their interactions with the platform will impact their continued usage intention. Therefore, this study proposes the following hypothesis:

 **H3.** *Customer-perceived emotional value* (**H3a**) *and social value* (**H3b**) *have a positive effect on customer stickiness.*

Sustainable consumption involves the behavior of obtaining, using, and disposing of resources in a manner that optimizes the social, economic, and environmental dimensions while meeting the needs of present and future generations [31,32]. In contrast to green consumption’s primary focus on environmental protection [33] and ethical consumption’s emphasis on fair practices by manufacturers and retailers [34,35], sustainable consumption embodies a more holistic comprehension of the intricate relationship between ethics, environment, and society [36,37].

Previous research has shown that initial green consumption often generates a “catalytic” effect, promoting individuals’ subsequent green consumption behavior [38]. After purchasing green products, consumers’ awareness and cognition of green products and environmental behavior are often enhanced. This enhanced awareness and cognition can expand to other areas, making it more likely for individuals to adopt subsequent green consumption behaviors. For instance, Sintov and colleagues discovered that the practice of composting household waste is conducive to fostering other energy-conserving behaviors within households [39]. Furthermore, Juhl et al. observed a certain degree of inertia in consumers’ purchasing habits of organic food [40]. In addition, scholars have found a cross-domain linkage phenomenon between online and offline green consumption behaviors [41]. By obtaining knowledge and information related to environmental protection through online channels, consumers can better understand environmental concepts and become more actively involved in offline environmental activities. For example, Zhang et al. found that users who frequently participate in Ant Forest activities online are more proactive in offline environmental activities [42]. Thus, it can be seen that the stickiness of using Ant Forest is beneficial for promoting offline sustainable consumption behaviors [43]. Therefore, this study proposes the following hypothesis:

 **H4.** 
*Customer stickiness has a positive effect on sustainable consumption behavior.*


Positive psychology suggests that pleasant experiences have the potential to facilitate various positive outcomes, including achievements in the workplace, pro-social behavior, and physical well-being. For instance, Zhang et al. found that pleasant experiences played a significant mediational role in connecting perceived benefits with consumers’ intention to purchase environmentally-friendly brands [44]. If individuals can obtain a satisfactory experiential value during their interactions, they are more inclined to participate in activities that generate value for businesses or platforms [45]. Customers who actively participate in Ant Forest’s activities and successfully plant trees are rewarded with official certificates and planting numbers. This strengthens customers’ recognition of Ant Forest’s value and fosters a greater sense of achievement. Recognizing the value and cultivating achievement motivation are vital factors that enhance customer loyalty [46]. The interface design of Ant Forest is user-friendly, facilitating seamless operation and usage, thereby increasing the probability of customers’ continuous engagement. The gamification design of Ant Forest deserves special mention as it enriches customers’ perception of emotional value, while the interactive design incorporating social attributes strengthens their perception of social value. The improvement in customers’ perception of Ant Forest’s value, in turn, reinforces their willingness to continue using Ant Forest, increasing their stickiness towards Ant Forest. Users’ ability to showcase their achievements through long-term environmental efforts fosters the ongoing development of sustainable environmental behavioral habits over a specific timeframe [47]. With prolonged usage, the travel habits of numerous users become more environmentally friendly and sustainable [48]. Based on the theory of planned behavior, customers’ online habits of engaging in green consumption can influence their offline behavior towards sustainable consumption. Therefore, this study proposes the following hypothesis:

 **H5.** *Passion prompts customers’ sustainable consumption behavior by indirectly influencing customer stickiness through perceived emotional value* (**H5a**) *and perceived social value* (**H5b**).

 **H6.** *Usability prompts customers’ sustainable consumption behavior by indirectly influencing customer stickiness through perceived emotional value* (**H6a**) *and perceived social value* (**H6b**).

Based on the analysis presented above, this study has developed a theoretical model that examines the influence of AI technology on sustainable consumption behavior by incorporating the S-O-R theory and the Theory of Planned Behavior. The proposed model, as shown in Figure 2, elucidates the impact of AI technology stimuli on customers’ sustainable consumption behavior, emphasizing the significance of customer-perceived value and customer stickiness in this course.

## 3. Materials and Method

### 3.1. Questionnaire Design

Based on the theoretical model, this study gathered data through a questionnaire survey consisting of five sections: demographic information (age, gender, occupation, education, and income), with four sections corresponding to the four theoretical variables. All research variables, excluding demographic information, were assessed utilizing the 7-point Likert Scale, which ranged from 7 (completely agree) to 1 (completely disagree). The variables, Passion, Usability, Emotional Value, Social Value, Customer Stickiness, and SCB, were characterized by assessing their mean values. Measurement items and their sources are listed in Table 1. Before officially distributing the questionnaire, 50 pre-survey questionnaires were distributed, and 42 valid responses were received. Scale validity was established by the Cronbach’s alpha value of all measurements exceeding 0.70.

### 3.2. Data Collection and Measurements

This study employed non-probability sampling methods to gather data. The formal questionnaire was primarily distributed through the Wenjuanxing platform, which is a reputable research platform in China. A total of 379 questionnaires were received, from which 99 invalid responses were discarded. The final 280 responses yielded an effective response rate of 73.9%. Specific demographic details of the participants are presented in Table 2. A valid questionnaire is defined as one that has been completed seriously by the target population, relying on the sample service. In particular, the seriousness of filling out the questionnaire was measured by examining the time spent and the source IP, ensuring the exclusion of respondents who completed the questionnaire casually or more than once. A diverse population of individuals with a range of ages, occupations, and educational backgrounds participated in the survey, with the majority of respondents holding an undergraduate degree or being students. It is important to note that younger users were more likely to use Ant Forest than their older counterparts. The resulting sample data reveal that the sample closely approximates real-world situations and is thus highly representative.

### 3.3. Data Analyses

This study applied the empirical research paradigm and a quantitative research method. Multivariate statistical analysis was conducted using SPSS 26.0 and AMOS 28.0 software, which included various methods such as exploratory factor analysis, confirmatory factor analysis, reliability analysis, regression analysis, and the Bootstrap method. Our goal was to evaluate the influence of artificial intelligence (AI) technology on sustainable consumption practices through customer-perceived value and customer stickiness.

## 4. Results

### 4.1. Reliability and Validity of the Measurement Instrument

SPSS 26.0 and AMOS 28.0 were utilized to evaluate the reliability and validity of the measurement model in this study. Both Cronbach’s alpha and Composite Reliability (CR) tests were performed to assess the structural reliability of the measurement model (see Table 3). The Cronbach’s alpha values ranged from 0.848 to 0.918, and both tests exceeded the threshold of 0.7 [55]. The convergent validity of the measurement model was tested by the Average Variance Extracted (AVE) and external factor loading [56]. Each respective factor loading was above 0.7 [55], and the AVE was above 0.5 [57], indicating high reliability and validity for each item. To test discriminant validity, both multi-factor and one-factor models were constructed (see Table 4). The fit indexes were lower than the measurement model, indicating good discriminant validity for both models. The Harman single-factor test was conducted to detect common method bias. The maximum single-factor variance was 24.955%, less than the threshold of 40%, indicating no serious common method bias in this study.

### 4.2. Model Fit

We assessed the model fit using AMOS (see Table 5). The results indicated a favorable fit between the data and the measurement model, as denoted by the chi-square degrees of freedom ratio (CMIN/DF) value of 2.216 (<3.0), the root-mean-square error of approximation (RMSEA) value of 0.066 (<0.08), the normed fit index (NFI) value of 0.915 (>0.9), the comparative fit index (CFI) value of 0.951 (>0.9), the incremental fit index (IFI) value of 0.951 (>0.9), and the Tucker-Lewis index (TLI) value of 0.943 (>0.9).

### 4.3. Hypotheses Testing

After conducting reliability and validity tests, the proposed hypotheses were tested using SPSS 26.0 software, and the main effect of each path in the model is shown in Figure 3. As indicated in Table 6, the two dimensions of AI technology stimuli, namely, “passion” and “usability”, had a significant positive impact on both emotional values (β = 0.675, *p* < 0.001; β = 0.664, *p* < 0.001) and social value (β = 0.371, *p* < 0.001; β = 0.378, *p* < 0.001). Hypotheses H1a, H1b, H2a, and H2b were thus verified. Furthermore, both dimensions of perceived customer value had a significant positive impact on customer stickiness (β = 0.732, *p* < 0.001; β = 0.552, *p* < 0.001), indicating that hypotheses H3a and H3b were supported. Furthermore, customer stickiness had a significant positive impact on sustainable consumption behavior (β = 0.528, *p* < 0.001), thereby supporting hypothesis H4.

### 4.4. Testing for Chain Mediation Effects

Building upon the path analysis, this study constructed four chain mediation models to examine the mediating effects of both customer-perceived value and customer stickiness. To estimate the parameters, the Bootstrap method of the SPSS 26.0 Process plugin was utilized, and a total of 5000 samples were taken. The results of the analysis are presented in Table 7, which shows the specific effect values.

Table 7 shows that the indirect effect of “passion → emotional value → customer stickiness → sustainable consumption behavior” is estimated to be 0.125 with a 95% Bootstrap confidence interval that excludes zero. This suggests that emotional value and customer stickiness have a mediational role between passion and sustainable consumption behavior, thus supporting hypothesis H5a. The indirect effect of “passion → social value → customer stickiness → sustainable consumption behavior” is estimated to be 0.043 with a 95% Bootstrap confidence interval that excludes zero. This suggests that social value and customer stickiness have a mediational role between passion and sustainable consumption behavior, thus providing support to hypothesis H5b. The indirect effect of “passion → social value → customer stickiness → sustainable consumption behavior” is estimated to be 0.043 with a 95% Bootstrap confidence interval that excludes zero. This suggests that social value and customer stickiness have a mediational role between passion and sustainable consumption behavior, thus providing support to hypothesis H5b. The indirect effect of “usability → emotional value → customer stickiness → sustainable consumption behavior” is estimated to be 0.106 with a 95% Bootstrap confidence interval that excludes zero. This suggests that emotional value and customer stickiness play a mediating role between usability and sustainable consumption behavior, providing evidence for hypothesis H6a. The indirect effect value of the path “usability → social value → customer stickiness → sustainable consumption behavior” was found to be 0.04 with a 95% Bootstrap confidence interval that did not include 0. This suggests that both social value and customer stickiness serve as mediators between usability and sustainable consumption behavior, therefore supporting hypothesis H6b.

A chain-mediated effect analysis revealed that the impact of artificial intelligence technology on promoting customer sustainable consumption behavior has intricate workings. The impact of artificial intelligence technology stimuli on sustainable consumption behavior through customer stickiness is less effective when compared to the impact of artificial intelligence technology stimuli on sustainable consumption behavior via perceived emotional value and customer stickiness. Nevertheless, the promotion of sustainable consumption behavior through customer stickiness influenced by perceived social value has a weaker effect when compared to directly influencing sustainable consumption behavior through customer stickiness. Therefore, research focused on promoting customers’ sustainable consumption behavior should pay more attention to customer-perceived emotional value rather than just customer-perceived social value.

## 5. Discussion and Conclusions

This study utilized the S-O-R theory and the TPB theory to analyze the impact mechanism of artificial intelligence technology stimuli on sustainable consumption behavior among customers using sample data from Ant Forest users. The research findings indicate that all six hypotheses have received validation. The specific conclusions are expounded as follows:

Firstly, artificial intelligence (AI) technology stimuli have a significant positive impact on customer-perceived value, which in turn significantly influences customer stickiness. The impact of passion and usability on customer-perceived emotional value (β = 0.675; β = 0.664) is greater than the impact on customer-perceived social value (β = 0.371; β = 0.378) when AI technology stimulates. When customer-perceived emotional value and customer-perceived social value act independently, customer-perceived emotional value has a greater impact on customer stickiness (β = 0.732) compared to customer-perceived social value (β = 0.552). This implies that prioritizing customer emotional value perception is crucial for fostering customer stickiness. Ant Forest users not only gain awareness of green consumption through the platform but also derive a sense of enjoyment and accomplishment. To maintain this feeling of achievement, customers are more inclined to transition their offline consumption behavior towards sustainable practices.

Secondly, the positive impact (β = 0.528) of customer stickiness on sustainable consumption behavior was found to be significant. The study further identified a “linkage effect” between online and offline sustainable consumption behaviors, where customers demonstrate their commitment to sustainable ideas and green lifestyles through consistent behaviors in both contexts. For instance, if a customer is a dedicated follower of Ant Forest, they are likely to be influenced by green consumption awareness when making purchases and prefer items with recyclable or reusable packaging. This preference aligns with their online purchasing habits. Hence, it can be observed that gamified green practice initiatives not only foster customer stickiness but also communicate the concept of green consumption to customers, thereby promoting sustainable consumption habits. Notably, these habits have the potential to transcend mere online purchases and permeate into various aspects of customers’ daily lives, expanding their influence beyond the realm of virtual transactions.

Finally, AI technology stimuli indirectly promote sustainable consumption behavior through customer-perceived value and customer stickiness. These factors, the impact of passion and usability on the adoption of sustainable consumer behavior, by influencing customer-perceived emotional value and fostering customer stickiness (β = 0.121; β = 0.100), outweigh the impact of passion and usability on sustainable consumer behavior via influencing customer-perceived social value and fostering customer stickiness (β = 0.043; β = 0.038). It further confirms that during the promotion of sustainable consumer behavior, customer-perceived emotional value holds greater importance compared to customer-perceived social value. Through gamified scenario design, AI technology can enhance the customers’ perception of hedonistic and utilitarian incentives, strengthening their emotional and social values and ultimately generating a sustained intention to use. Due to the “linkage effect” between online and offline consumption behaviors, external environmental stimuli perceived by customers, such as AI technology, can lead to consistent consumption behaviors. This study applied the S-O-R and TPB theories to integrate AI technology stimuli with customer-perceived value, customer stickiness, and sustainable consumption behavior using Ant Forest as an example. The study provides a comprehensive and systematic explanation of the impact mechanism of AI technology stimuli on sustainable consumption behavior.

### 5.1. Theoretical and Practical Significance

The theoretical significance of this study lies in the integration of the S-O-R and TPB to propose a model that demonstrates the significant promoting effect of AI technology stimuli on sustainable consumption behavior and elucidates the “linkage effect” between online and offline platforms. The survey results from 280 respondents provide empirical support for the theoretical significance of this model. This study introduces a novel approach to encourage sustainable consumption behavior by leveraging AI technology. This strategy focuses on influencing customers’ perceived value, stimulating their involvement, and enhancing their stickiness. As a result, this study offers theoretical inspiration for exploring the factors influencing sustainable consumption behavior from the perspective of artificial intelligence.

This research holds practical significance in four key aspects: customers, managers, organizations, and society. For customers, Ant Forest combines gamification and emotional design to reduce the barriers to using environmental protection apps. This enables users to engage in environmental causes through actions that are within their capabilities. Moreover, it provides users with a sense of satisfaction and accomplishment while also fostering their awareness of carbon reduction through gameplay. Moreover, the successful integration of virtual and real-life elements effectively conveys the concept of low-carbon living to the general public, subtly enhancing users’ behavioral habits through gameplay, and converting them into active participants and advocates for green environmental causes, ultimately attaining self-fulfillment. For managers, the adoption of sustainable strategies is motivated by sustainable consumption behavior, which enhances the organization’s social reputation and capacity for sustainable development. Managers have the ability to create environmental and social responsibility policies, oversee their implementation, and foster sustainable growth and innovation within the organization. Moreover, leveraging artificial intelligence technology can effectively guide customers towards sustainable consumption behavior and cultivate their loyalty. For this reason, managers should prioritize its utilization. For organizations, with the increasing importance consumers place on environmental issues and social responsibility, there has been a corresponding rise in the demand for sustainable products and services. Organizations should concentrate on enhancing the perceived emotional value of customers by developing high-quality and sustainable consumer products and services that cater to their needs, thus fostering customer loyalty. Simultaneously, organizations have the opportunity to establish corporate values surrounding sustainable consumption and cultivate a socially responsible corporate image. These efforts prove instrumental in gaining consumer trust and loyalty, as well as attracting a broader customer and partner base. For society, the sustainable consumption behavior of customers plays a crucial role in advancing society’s goals of sustainable development, encompassing environmental protection, social equity, and economic prosperity. By actively encouraging customer engagement in sustainable consumption practices, society can effectively utilize resources, protect the environment, and promote social equality and inclusivity. The promotion of sustainable consumption behavior among customers also serves to enhance societal awareness and recognition of the significance of sustainability. Additionally, it fosters social innovation and cooperation, motivating all stakeholders to actively participate in sustainable development and collectively shape a more sustainable future society.

### 5.2. Limitations and Future Directions

In the era of rapid technological developments, including big data, the impact of AI technology stimuli is likely to be more extensive, necessitating further research. Firstly, the study had a limited sample size. Future research should examine the consistency of conclusions when using a larger sample size. Future research should also employ more advanced research methods, such as big data analysis, to achieve a more comprehensive sample and increase the accuracy of the results. In addition, future research should employ more advanced research methods, such as big data analysis, to achieve a more comprehensive sample and increase the accuracy of the results. Secondly, the variable dimensions were limited. This study only evaluated the influence of two dimensions of AI technology stimuli (passion and usability) on selected variables (customer perception, intention, and behavior), without fully considering other variables, such as customer values and technological readiness. Thus, it is necessary for future research to explore additional dimensions of AI technology stimuli, as well as to consider larger data sizes and more diverse case samples. Moreover, variables related to customer self-factors (values, etc.) should be taken into account while examining the impact of external factors.

## Figures and Tables

**Figure 1 behavsci-13-00604-f001:**
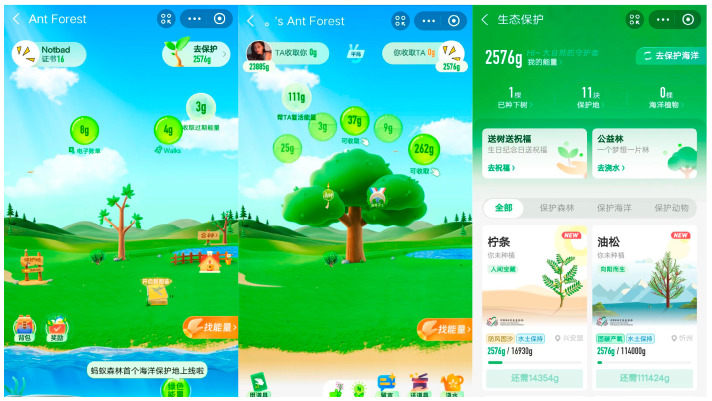
Partial functions of Ant Forest.

**Figure 2 behavsci-13-00604-f002:**
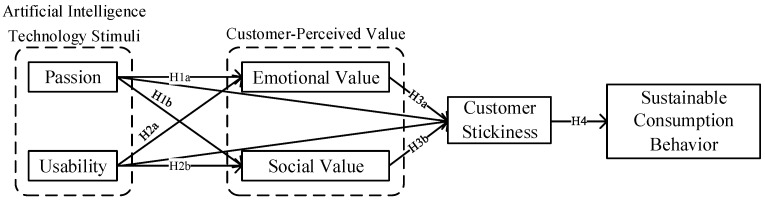
The theoretical model of the study.

**Figure 3 behavsci-13-00604-f003:**
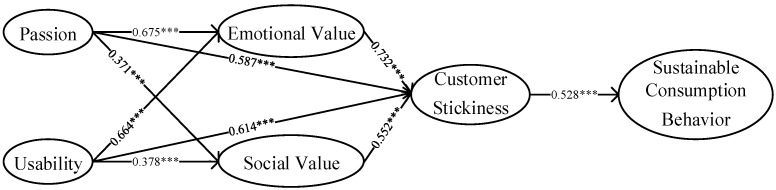
Path coefficient diagram. Note: *** indicates *p* < 0.001.

**Table 1 behavsci-13-00604-t001:** Summary of the Construct Items and Sources.

Index	Code	Final Item	Source
Passion (P)	P1	I would like to use Ant Forest because I am passionate about it.	Brian et al., 2015 [49]
	P2	I use Ant Forest because I am passionate about protecting the environment.	
	P3	When I plant a tree through Ant Forest, I feel excited.	
Usability (U)	U1	I think the content and structure of Ant Forest is easy to understand.	Carlos et al., 2005 [50]
	U2	I think Ant Forest is easy to play, even for the first time.	
	U3	When I use Ant Forest, I feel I can control what I do.	
Emotional Value (EV)	EV1	The process of playing Ant Forest is enjoyable.	Zhang et al., 2020 [51] & Guo et al., 2023 [38]
	EV2	Ant Forest has brought me a lot of happiness.	
	EV3	Overall, I think Ant Forest is fun.	
	EV4	Using Ant Forest makes me feel excited.	
Social Value (SV)	SV1	I keep in touch with my friends through Ant Forest.	Shi et al., 2022 [52] & Zhou et al., 2021 [53]
	SV2	I have met new people through Ant Forest.	
	SV3	Ant Forest has helped me improve my social relationships.	
	SV4	Through Ant Forest, it has made it easier for me to gain the approval of others.	
Customer Stickiness (CS)	CS1	I plan to continue playing Ant Forest in the future.	Zhang et al., 2020 [51]
	CS2	I will play Ant Forest as often as I do now.	
	CS3	I will play Ant Forest as much as possible.	
Sustainable Consumption Behavior (SCB)	SCB1	I perform daily activities to care for and preserve the environment.	Muhammad et al., 2020 [54] & Huang et al., 2023 [14]
	SCB2	I would like to make changes in my lifestyle in search of more responsible consumption.	
	SCB3	I am satisfied with my responsible consumption behaviors.	
	SCB4	I purchase and use environmentally friendly products.	

**Table 2 behavsci-13-00604-t002:** Respondents’ Demographic Profiles.

		Total N = 280	
		Frequency	%
Gender	Male	114	40.71
	Female	166	59.29
Age	Below 25	88	31.43
	25–34	105	37.50
	35–49	53	18.93
	50–64	28	10.00
	Above 64	6	2.14
Occupation	Student	95	33.93
	Office worker	58	20.71
	Technician	37	13.21
	Professionals	43	15.36
	Individual operator	18	6.43
	Researcher	12	4.29
	Scholar	3	1.07
	Other occupation	14	5.00
Education	Below Undergraduate	58	20.71
	Undergraduate	165	58.93
	Postgraduate	57	20.36
Monthly income	Below CNY 5000	129	46.07
	CNY 5000–10,000	62	22.14
	CNY 10,001–15,000	53	18.93
	CNY 15,001–20,000	27	9.64
	Above CNY 20,000	9	3.22

**Table 3 behavsci-13-00604-t003:** The Measurement Model Assessment Results.

Construct	Item	Mean	Standard Dev.	Standardized Loading	Cronbach’s Alpha	CR	AVE
Passion	P1	5.16	1.447	0.821	0.848	0.849	0.652
	P2	5.22	1.383	0.793			
	P3	5.41	1.290	0.808			
Usability	U1	5.38	1.278	0.819	0.854	0.857	0.666
	U2	5.31	1.510	0.841			
	U3	5.19	1.428	0.788			
Emotional Value	EV1	5.25	1.268	0.817	0.892	0.893	0.676
	EV2	5.18	1.365	0.848			
	EV3	5.26	1.440	0.825			
	EV4	5.05	1.389	0.799			
Social Value	SV1	4.91	1.492	0.802	0.918	0.919	0.740
	SV2	4.56	1.625	0.870			
	SV3	4.64	1.722	0.904			
	SV4	4.53	1.659	0.861			
Customer Stickiness	CS1	5.15	1.414	0.834	0.864	0.865	0.682
	CS2	5.07	1.261	0.823			
	CS3	5.09	1.326	0.820			
Sustainable Consumption Behavior	SCB1	5.40	1.319	0.847	0.904	0.904	0.702
	SCB2	5.29	1.338	0.844			
	SCB3	5.29	1.297	0.819			
	SCB4	5.39	1.347	0.842			

**Table 4 behavsci-13-00604-t004:** Discriminant Validity Testing.

Model	CMIN	DF	CMIN/DF	NFI	CFI	RMSEA
Measurement model	370.877	174	2.131	0.921	0.956	0.064
Five-factor model	683.872	179	3.821	0.853	0.887	0.101
Four-factor model	1104.080	183	6.033	0.763	0.793	0.134
Three-factor model	1110.708	186	5.972	0.762	0.792	0.133
Two-factor model	1408.872	188	7.494	0.698	0.726	0.153
One-factor model	1669.997	189	8.836	0.642	0.668	0.168

Note: Five-factor model: P, U, EV, SV, CS + SCB; Four-factor model: P, U, EV + SV, CS + SCB; Three-factor model: P + U, EV + SV, CS + SCB; Two-factor model: P + U, EV + SV + CS + SCB; One-factor model: P + U+EV + SV + CS + SCB.

**Table 5 behavsci-13-00604-t005:** Measures of the model fit.

Fit Index	CMIN/DF	RMSEA	NFI	CFI	IFI	TLI
Model value	2.216	0.066	0.915	0.951	0.951	0.943

**Table 6 behavsci-13-00604-t006:** Hypotheses Testing Results.

Relationship	β	SE	t	R^2^
Passion → Emotional Value	0.675	0.041	16.059 ***	0.543
Usability → Emotional Value	0.664	0.040	15.876 ***	0.538
Passion → Social Value	0.371	0.060	7.487 ***	0.364
Usability → Social Value	0.378	0.057	7.731 ***	0.371
Passion → Customer Stickiness	0.587	0.049	11.876 ***	0.368
Usability → Customer Stickiness	0.614	0.045	12.925 ***	0.406
Emotional Value → Customer Stickiness	0.732	0.044	16.438 ***	0.519
Social Value → Customer Stickiness	0.552	0.048	9.386 ***	0.276
Customer Stickiness → Sustainable Consumption Behavior	0.528	0.048	10.742 ***	0.369

Note: *** indicates *p* < 0.001.

**Table 7 behavsci-13-00604-t007:** The results of chain mediation analysis.

Relationship	Mediation Effect	Proportion of Effect	95%CIs
Ind1 Passion → Emotional Value → Sustainable Consumption Behavior	0.000		[−0.041, 0.219]
Ind2 Passion → Customer Stickiness → Sustainable Consumption Behavior	0.054	30.66%	[0.008, 0.111]
Ind3 Passion → Emotional Value → Customer Stickiness → Sustainable Consumption Behavior	0.121	69.34%	[0.042, 0.208]
Total Mediation Effect: Ind1+ Ind2+ Ind3	0.175	100%	[0.149,0.377]
Ind4 Passion → Social Value → Sustainable Consumption Behavior	0.000		[−0.034, 0.066]
Ind5 Passion → Customer Stickiness →Sustainable Consumption Behavior	0.157	78.43%	[0.079, 0.245]
Ind6 Passion → Social Value → Customer Stickiness → Sustainable Consumption Behavior	0.043	21.57%	[0.017, 0.080]
Total Mediation Effect: Ind4+ Ind5+ Ind6	0.200	100%	[0.123, 0.315]
Ind7 Usability → Emotional Value → Sustainable Consumption Behavior	0.000		[−0.033, 0.218]
Ind8 Usability → Customer Stickiness → Sustainable Consumption Behavior	0.066	39.81%	[0.016, 0.129]
Ind9 Usability → Emotional Value → Customer Stickiness → Sustainable Consumption Behavior	0.100	60.19%	[0.031, 0.177]
Total Mediation Effect: Ind7+ Ind8+ Ind9	0.166	100%	[0.156, 0.361]
Ind10 Usability → Social Value → Sustainable Consumption Behavior	0.000		[−0.035, 0.067]
Ind11 Usability → Customer Stickiness → Sustainable Consumption Behavior	0.156	80.47%	[0.079, 0.243]
Ind12 Usability → Social Value → Customer Stickiness → Sustainable Consumption Behavior	0.038	19.53%	[0.015, 0.070]
Total Mediation Effect: Ind10+ Ind11+ Ind12	0.194	100%	[0.123, 0.304]

## Data Availability

Data available on request due to restrictions eg privacy or ethical. The data presented in this study are available on request from the corresponding author. The data are not publicly available due to the more private nature of the questionnaire data.

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
