# Peer review of "The Impact of Artificial Intelligence Technology Stimuli on Sustainable Consumption Behavior: Evidence from Ant Forest Users in China"

_behavsci, 2023, doi:10.3390/bs13070604_

Round 1

Reviewer 1 Report

This article investigates how people’s attitudes toward the app “Ant Forest” influence customer retention and evaluation of the app. The main problem with the study is that the effect is obvious. You technically predict that people will use and like the app more if they feel passionate about the app and find it easy to use the app.

However, “Ant Forest” is interesting. I just did a quick Google search. A mobile app that tracks a person’s carbon footprint and helps them reduce carbon emissions is something very new to people outside China. I recommend that the authors reorganize this paper into a case study of “Ant Forest,” give a detailed review of the motivation and history behind this campaign.

Only hypotheses 4,5, and 6 make an authentic contribution. Focus on these three hypotheses. Expand the introduction by focusing on the case study of “Ant Forest.” Rewrite the abstract and the title. Add figures that show how this app functions.

This article does not have a language issue. 

Author Response

Dear Reviewer:

Thank you for your time and effort in providing valuable comments on our article, the revised manuscript has now been uploaded according to your suggestions and requirements, and the responses to your suggestions are explained in detail in the attached document.
Thank you again for your time and valuable comments.

Sincerely

Reviewer 2 Report

The topic is interesting, but the article needs some improvements in order to be approved for publishing.

Thus, in the Abstract, the authors must add, besides general results, some data to indicated what it is been said about that. At the final part of the Abstract, you need to add a few proposals for improvement, future directions for reserach.

In the Introduction, you must add about its novelty, the goal, and at its final part a short presentation of each chapter. Here, you also must take all the info about the model building and move it to the third part, before 3.1. and than to present the questionnaire.

At subject 2.2. for each developed hypothesis, you need more bibliographic sources, in order to be established like that. You only have 2 sources per hypothesis, and using only two sources, you establish two research hypoyheses. Are indicated materials and sources separatelly for each hypothesis. And because you have very few sources (only 33) you need to add much more. An article, based on sources, must use a lot more than 33 sources. At least double, so 60 maiby 70 sources are indicated for a good article.

The methodology is good.

At 5. Section, you need to accompany the indicated results with data obtained (values), and more specifically discussed and made a better interpretation.

At 5.1. subject, you need to make the proposals and significance on four categories- for customers, for managers, for organizations and for society. So, this section needs serious improvements.

As it was already added, the References need important improvements- there is a reduced number of sources, so you need to increase it, and more importantly is that you used only two sources from 2023, so, if you must add new sources, is indicated to add them updated, too.

So, the article needs major improvements, in order to be approved for publishing.

Author Response

(The authors gave the same response as above.)

Round 2

Reviewer 1 Report

The authors addressed all my concerns.

This article is easy to follow.

Reviewer 2 Report

The authors made the suggested demands in order to improve the content of the article.

Now, in this form, the article is prepared for publication.